# Prospective Evaluation of Pulse Oximetry Screening for Critical Congenital Heart Disease in a Jordanian Tertiary Hospital: High Incidence and Early Detection Challenges

**DOI:** 10.3390/pediatric17010023

**Published:** 2025-02-15

**Authors:** Naser Aldain A. Abu Lehyah, Abeer A. Hasan, Mahmoud Y. Abbad, Razan A. Al-Jammal, Moath K. Al Tarawneh, Dima Abu Nasrieh, Haneen A. Banihani, Saif N. Aburumman, Areen G. Fraijat, Heba M. Alhawamdeh, Qasem A. Shersheer, Milad Kh. Al-Awawdeh, Scott O. Guthrie, Joseph R. Starnes

**Affiliations:** 1Division of Neonatology, Department of Pediatrics, Women’s and Children’s Hospital, Al Bashir Hospitals, Amman 11151, Jordan; nasser_abulehia@yahoo.com (N.A.A.A.L.); abeerjammaldr@yahoo.com (A.A.H.); 2Department of General Pediatrics, Women’s and Children’s Hospital, Al Bashir Hospitals, Amman 11151, Jordan; dr.mabbad@yahoo.com (M.Y.A.); moath.tarawneh1985@gmail.com (M.K.A.T.); shersheer67qasem@gmail.com (Q.A.S.); 3Department of Pediatrics, King Abdullah University Hospital, Ar-Ramtha 22110, Jordan; razanjammal1999@hotmail.com; 4School of Medicine, University of Jordan, Amman 11942, Jordan; dimanasrieh@gmail.com (D.A.N.); aburummansaif@yahoo.com (S.N.A.); 5Department of General Pediatrics, Al Tafilah Governmental Hospital, Al Tafilah 64710, Jordan; areenfraijat2@yahoo.com; 6Department of Obstetrics and Gynecology, Women’s and Children’s Hospital, Al Bashir Hospitals, Amman 11151, Jordan; omomarnassa@gmail.com; 7Al Bashir Hospitals Administration, Amman 11151, Jordan; milad.alawawdeh@ehs.com.jo; 8Electronic Health Solutions (EHS), Amman 11151, Jordan; 9Division of Neonatology, Department of Pediatrics, Vanderbilt University Medical Center, Nashville, TN 37232, USA; scott.o.guthrie@vumc.org; 10Division of Pediatric Cardiology, Department of Pediatrics, Vanderbilt University Medical Center, Nashville, TN 37232, USA

**Keywords:** pulse oximetry, screening, critical congenital heart disease screening, neonatal screening, echocardiography

## Abstract

Background/Objectives: Critical congenital heart disease (CCHD) is among the major causes of global neonatal morbidity and mortality. While the incidence of CCHD appears to vary across populations, much of this variation may stem from differences in detection and reporting capabilities rather than true prevalence. In Jordan, recent data revealed a congenital cardiac disease incidence of 17.8/1000 live births, much higher than international averages. Diagnosis is largely dependent upon echocardiography, which is difficult to obtain in low-resource settings where prenatal screening modalities are limited. Screening for CCHD with pulse oximetry offers a potential method to identify patients earlier and contribute to improved outcomes. Methods: This prospective cohort study evaluated 20,482 neonates screened using pulse oximetry at Al-Bashir Hospital between January 2022 and May 2024. Demographic data, pulse oximetry measurements, and echocardiogram findings were collected during the screening process after obtaining ethical approval from the Jordanian Ministry of Health. Results: Pulse oximetry screening identified 752 neonates (3.7%) requiring further evaluation by echocardiography. An abnormality was detected in 240 neonates (31.9%), which included cardiac anomalies and pulmonary hypertension. Screening led to the identification of 138 infants with CCHD, including 80 with a previously unknown diagnosis, and an additional 247 infants with conditions requiring increased monitoring or treatment. Among those with CCHD, hypoplastic left heart syndrome and Tetralogy of Fallot were the most common conditions, 3.1%, and 2.4%, respectively. The overall false positive rate was 1.8% and was higher among those screened at less than 24 h of life compared to those screened at or after 24 h of life (2.3% [95%CI 2.1–2.6] vs. 0.8% [95%CI 0.6–1.0], *p* < 0.001). Conclusions: Pulse oximetry screening successfully led to the early detection of CCHD among Jordanian neonates. There was a high prevalence of CCHD compared to other reported cohorts. This highlights the importance of implementing national screening protocols to improve early diagnosis and intervention. Future studies will inform the feasibility and cost-effectiveness of national implementation in this setting.

## 1. Introduction

Congenital heart disease (CHD) is the most common birth defect worldwide, affecting 8 to 12 per 1000 live births [1,2]. CHD is the leading cause of non-communicable death among individuals less than 20 years of age and causes more than 250,000 deaths globally each year, with more than 180,000 occurring in infants [3,4]. According to a study conducted at Al-Bashir Hospital in Jordan between 2019 and 2021, which accounted for about 10% of all deliveries in Jordan and 20% of all deliveries in Ministry of Health hospitals, the incidence of cardiac disease was notably higher at 17.8 per 1000 live births, excluding patent ductus arteriosus (PDA) [5]. PDA was the most commonly identified acyanotic congenital heart defect, identified in 27.4% of preterm babies born with CHD, and increased the incidence of cardiac disease to 24.6 per 1000 live births [5].

Critical congenital heart disease (CCHD) is a subset of CHD generally defined as CHD requiring intervention in the first year of life for survival [6]. Definitive diagnosis of CHD and CCHD requires cardiac imaging modalities, most commonly an echocardiogram [7]. To diagnose CHD prenatally, a fetal echocardiogram is needed. Although this is increasingly common in high-resource settings, it is uncommon in low-resource settings in middle-income countries as access to fetal echocardiography is rare [8].

Pulse oximetry is a ubiquitous patient monitoring tool used across diverse settings as a non-invasive proxy for arterial oxygen saturation. Pulse oximetry works by measuring the transmission or reflection of a light source, and a saturation can be obtained using the differential absorption of light by oxygenated and deoxygenated hemoglobin. Using pulse oximetry screening (POS) to screen neonates for CCHD has been studied in various countries including China [9], the United States of America [10], India [11], Tanzania [12], and Indonesia [13]. The main aim of screening is to achieve early diagnosis of patients with CCHD when not detected prenatally. According to a meta-analysis conducted in 2018 and involving 21 studies (N = 457,202 participants), using pulse oximetry as a screening method for CCHD has a sensitivity and specificity of 76.3% and 99.9%, respectively, with a false positive rate of just 0.1% [14]. Importantly, pulse oximetry has the added benefit of identifying other disease states requiring intervention, such as pneumonia and sepsis. This has been seen in studies in a variety of global settings [15,16,17].

Pulse oximetry screening for CCHD has been implemented as a mandatory screening method in numerous high-income countries [18]. Implementation in low- and low–middle-income countries (LLMICs), however, has been harder, which results in delayed diagnosis and treatment of CCHD. There is currently no national CCHD screening policy in Jordan. Delayed diagnosis may contribute to worse patient outcomes [19].

As a tertiary referral center, Al-Bashir hospital delivers almost 15,000 babies each year. This, coupled with the high rates of CCHD in the population, stresses the importance of using a quick, effective, and accurate screening tool to diagnose and treat CCHD early. This need is further emphasized by low prenatal detection rates. Although fetal echocardiography is available at some referral centers in Jordan, prenatal detection rates remain very low, even at tertiary and quaternary hospitals [5]. Improved early detection may in turn improve mortality rates, helping achieve the World Health Organization’s Sustainable Development Goals (SDG) 3.2 and 3.4 aimed at reducing neonatal mortality and premature deaths from non-communicable diseases [20].

This study is the first performance report of its kind in Jordan, conducted to assess the reliability and accuracy of using POS to screen neonates for CCHD. Previous studies of CCHD incidence in Jordan have reported on infants identified clinically, and no previous studies have reported infants identified by pulse oximetry screening [5].

## 2. Materials and Methods

### 2.1. Study Design and Participants

This is a prospective cohort study of all newborns admitted to the newborn nursery at Al-Bashir Hospital between January 2022 and May 2024, excluding those admitted to the NICU and who left against medical advice.

Infants diagnosed with CCHD prenatally were screened postnatally with POS and an echocardiogram to confirm the diagnosis. This step was essential to verify the presence of CCHD as prenatal diagnoses were made by general obstetricians and not fetomaternal specialists. Accordingly, screening was necessary to help confirm the presence and type of CCHD, facilitating appropriate intervention if required.

Infants immediately admitted to the NICU following birth and with no initial prenatal diagnosis of CCHD were excluded. This included preterm infants (<35 weeks gestational age). This is because all neonates in the NICU require respiratory support and supplemental oxygen, which influences the POS reading, making it challenging to differentiate CCHD from respiratory conditions. By excluding these cases, we aim to maintain the accuracy and reliability of the screening process for detecting undiagnosed CCHD in stable, term, or near-term infants.

Of 23,796 infants admitted to the nursery, 20,482 (86.0%) underwent screening. The remaining 3314 (13.9%) left the hospital early against medical advice and did not have the opportunity to be screened. These infants were excluded from this analysis.

After obtaining approval from the Jordanian Ministry of Health’s Institutional Ethics Committee, the electronic medical records of the screened neonates were reviewed. Demographic information, pulse oximetry measurements, and echocardiogram findings were evaluated. Consanguinity was defined as marriage between second-degree relatives, including first cousins.

### 2.2. Definitions

#### 2.2.1. Critical Congenital Heart Disease

CCHD was defined using a similar definition to that proposed in previous systematic reviews [1]. Specifically, this was defined as a life-threatening, ductal-dependent lesion leading to death or requiring an interventional procedure in the first 28 days of life. This included hypoplastic left heart syndrome, pulmonary atresia, transposition of the great arteries, interrupted aortic arch, coarctation of the aorta, critical aortic stenosis, critical pulmonary stenosis, Tetralogy of Fallot, total anomalous pulmonary venous return, and any other disease requiring intervention in the first 28 days of life for survival [14]. Non-critical CHDs, such as small ventricular septal defects or patent ductus arteriosus (PDA), were excluded from this analysis.

#### 2.2.2. Persistent Pulmonary Hypertension of the Newborn (PPHN)

PPHN is a syndrome caused by elevated pulmonary vascular resistance, leading to labile hypoxemia due to decreased pulmonary blood flow and right-to-left shunting of blood [21]. We defined PPHN similarly to prior pulse oximetry studies [22]:Pulse oximetry results: Significant pre- and post-ductal oxygen saturation differences (≥10%) on pulse oximetry screening.Echocardiographic findings: Elevated pulmonary artery pressures as evidenced by significant tricuspid regurgitation and right-to-left shunting across the PDA or foramen ovale.

This combination of clinical and echocardiographic criteria ensured accurate identification of PPHN cases. The definition and diagnostic approach align with previously established criteria in related studies.

#### 2.2.3. Infections

Neonatal sepsis was defined as raised inflammatory markers (C-reactive protein [CRP] > 10 mg/dL) with or without a positive blood culture, requiring antibiotics for at least five days [17]. Congenital pneumonia was defined as respiratory distress with raised inflammatory markers (CRP > 10 mg/dL) and/or a positive blood culture, along with radiological changes on a chest X-ray. Treatment required antibiotics for a minimum of five days and oxygen support for at least two hours [17].

### 2.3. Study Setting

Al-Bashir Hospital, located in the capital city of Amman, is the largest hospital in Jordan. It is composed of five separate hospitals, has many residency programs, and serves as a teaching hospital for multiple universities. One of these five hospitals is the Women and Children’s Hospital, which is a tertiary referral hospital receiving referrals from throughout the country. Pulse oximetry screening for CCHD at Al-Bashir began in 2020 and has since become mandatory for babies delivered at the hospital. Patients requiring cardiac surgery are referred to Queen Alia Heart Institute for their procedures.

### 2.4. Procedure

All screened newborns underwent a minimum of one POS and a maximum of two. If the infant passed based on the American Academy of Pediatrics (AAP) protocol [23], they were provided normal newborn care and did not have to undergo any additional testing. If they failed both the first and second measurements, they were not tested further and were referred to a senior pediatric cardiologist for immediate assessment for CCHD using echocardiography. The screening protocol used is outlined in detail in Figure 1. At the time of data collection, this was a proposed algorithm by the AAP [23], which has since become the new algorithm formally recommended by the AAP [24].

#### 2.4.1. Pulse Oximetry

Pediatric residents, interns, and nursing staff from the obstetrics department performed pulse oximetry measurements using a reusable pulse oximetry probe. Diagrams describing screening guidelines were displayed in both the postpartum and NICU departments. Monthly meetings were held to reinforce screening processes and to review cases identified by screening.

All babies were screened 24 h after birth or earlier if being discharged. Pulse oximeter oxygen saturation (SpO2) was measured from the right hand and on either foot, and SpO2 levels were recorded. Results were interpreted and managed according to the protocol above (Figure 1).

#### 2.4.2. Echocardiography

If a newborn failed two pulse oximetry measurements, they were referred to a senior pediatric cardiologist to undergo echocardiogram with 2D images as well as color and spectral Doppler assessment to determine the presence of a CCHD. The Vivid T8 ultrasound system (GE HealthCare, Chicago, IL, USA) was used for echocardiography. A detailed echocardiogram was performed to evaluate cardiac anatomy, systemic and pulmonary veins, and valve function, allowing the pediatric cardiologist to determine the presence of CHD based on the International Pediatric and Congenital Cardiac Code [25].

### 2.5. Data Analysis

Data analysis was conducted using IBM SPSS software version 26.0 (IBM, Chicago, IL, USA) and Microsoft Excel 365 (Microsoft, Redmond, WA, USA). Mean (M) and standard deviation (SD) or median (Md) and interquartile range (IQR) were used to describe continuous variables (e.g., age), and frequencies and percentages were used to summarize categorical data. Independent-sample t-Tests were used to examine differences in means. Chi-Square Tests of Independence were performed to assess the relationship between categorical variables. Statistical significance was defined as a *p*-value less than 0.05.

## 3. Results

### 3.1. Demographics

A total of 20,482 neonates from Al Bashir Hospital were screened between January 2022 and May 2024 and were part of the study, as depicted in Table 1. There were 11,883 males (58.0%) and 8599 females (42.0%). Neonates had an average gestational age at delivery of 38.7 ± 1.5 weeks and a birth body weight of 3.2 ± 0.7 kg. A high prevalence of second-degree consanguinity was reported in 6207 neonates (30.3%). Most neonates were Jordanians (*n* = 18,404, 89.9%). Only 44 neonates had a family history of CCHD (0.2%).

### 3.2. Results of Screening

The results of screening and echocardiography are illustrated in Figure 2. The first SpO2 measurement was taken at a median time of 20 h after birth (IQR, 15 to 24). A total of 7259 infants (35.4%) were screened ≥24 h from birth. The majority (19,467, 95.0%) passed the first measurement, and 263 (1.3%) passed the second measurement. A total of 752 neonates failed the screening test and underwent echocardiography (3.7%).

Among those who underwent echocardiography, 31.9% had abnormal findings (*n* = 240), including PPHN. Cardiac anomalies were the most common (57.5%), followed by PPHN (42.5%). Hypoplastic left heart syndrome was the most common cardiac anomaly (*n* = 23, 3.1%) followed by Tetralogy of Fallot (*n* = 18, 2.4%), as depicted in Table 2. Among patients with cardiac anomalies, 50% had consanguineous parents (*n* = 69) and 30.4% had a family history of CHD (*n* = 42). The incidence of cardiac anomalies was lower in Jordanians (*n* = 95, 0.5%) compared to non-Jordanians (*n* = 43, 2.1%) (*p* < 0.001). Cardiac anomalies were more common in infants with history of consanguinity (*n* = 69, 1.1%) compared to infants without history of consanguinity (*n* = 69, 0.5%), (*p* < 0.001). Only 58 neonates had prenatal diagnoses of cardiac anomalies.

### 3.3. Screening Test Performance

A total of 20,097 infants (98.1%) were healthy newborns. Cardiac anomalies were diagnosed in 138 neonates (0.7%). The overall incidence of CCHD in the population was 6.6 per 1000 live births. PPHN had an overall incidence of 0.5% (*n* = 102); 0.4% of neonates were septic (*n* = 85), and 0.3% had congenital pneumonia (*n* = 60). The false-positive rate for children who screened positive and did not have CCHD was 3.0%. When including other non-CCHD pathologies (PPHN, pneumonia, sepsis), this dropped to 1.8%. Similarly, the specificity considering all children without CCHD as false positives was 97.0%. This increased to 98.2% when considering only those who had no ultimate diagnosis as false positives.

False-positive neonates were screened at a mean of 12.0 ± 6.5 h after birth, as compared to 19.7 ± 6.9 h for the true-positive neonates (*p* < 0.001). False-positive rates for infants screened less than 24 h after birth were significantly higher than those screened ≥ 24 h after birth (*p* < 0.001), as depicted in Table 3.

## 4. Discussion

This study is the first of its kind to evaluate pulse oximetry screening for CCHD in Jordan and is one of the largest screening studies in a middle-income country. It offers valuable insights into the burden, early identification, and handling of this illness in a middle-income country. The screening was successfully integrated into a referral hospital’s operations, resulting in a very high number of infants being screened. This led to the detection of 138 babies with CCHD, including 80 who had not been diagnosed earlier, and an additional 247 babies with diseases requiring additional monitoring or treatment.

The incidence of CCHD in our cohort was found to be high at 6.6 per 1000 live births, significantly higher than the international prevalence of approximately 2 per 1000 [26,27,28]. A high incidence of consanguinity and other environmental issues, including maternal smoking, exposure to teratogenic drugs, nutritional deficiencies, and hypoxia, may contribute to the incidence of congenital heart disease in Jordan [29,30]. This may also reflect the tertiary setting. A study across Europe involving fifteen birth defect surveillance programs showed a median total incidence of CCHD of 1.9 per 1000, with a range from 1.0 to 3.1 per 1000 live births [26]. Another study on severe non-chromosomal CHD, which included CCHD, conducted between 2000 and 2005 in Europe estimated the accumulation rate at 2 cases per 1000 births [27]. Similarly, a long-term study in Atlanta found an approximate incidence rate of 1.7 for every 1000 live births [28]. The prevalence of specific cardiac anomalies identified in our study was comparable with other regional reports [9,10,11,12,13]. Specifically, hypoplastic left heart syndrome and Tetralogy of Fallot were the most common lesions identified.

The specificity for POS in our study was 97–98% depending on the definition of a false positive. This is lower than the 99.9% found in large meta-analyses [14]. This may be due to recent changes in the recommended screening algorithm, which included removing a second retest [23,24]. It also likely reflects that many of the infants in this study were screened prior to 24 h of life because they were either being discharged or were being taken home early against medical advice. The literature shows that testing prior to 24 h of life leads to lower specificity [31]. Differences in discharge timing, including parental expectations, must be considered when designing large screening programs.

Early screening often leads to higher false-positive rates due to the natural physiological changes in oxygen saturation that occur shortly after birth. This finding underscores the importance of timing in screening protocols. The rationale for conducting early screening in our study was driven by the need to identify CCHD before hospital discharge, especially since a significant number of families leave the hospital against medical advice. To overcome the problem of early discharge and improve the accuracy of CCHD screening, future steps will include educating parents regarding the necessity of staying in the hospital for the recommended period of observation and follow-up screenings at local healthcare centers if early discharge cannot be avoided.

In addition to identifying CCHD, our study highlighted the secondary benefit of pulse oximetry screening in detecting sepsis and pneumonia in newborns. Pulse oximetry screening is well documented in the literature as a useful tool for early identification of these conditions [15,16,17]. Infants with hypoxemia were identified through monitoring their oxygen saturation levels, which necessitated conducting additional evaluations leading to earlier diagnosis of sepsis and pneumonia. Hence, besides helping in the early detection of critical heart defects, this dual advantage of pulse oximetry screening improves general neonatal care, thus enabling prompt handling of other serious conditions.

Further studies are needed to elucidate the long-term outcomes for infants with a diagnosis of congenital heart disease by pulse oximetry screening and to assess the large-scale cost-effectiveness of pulse oximetry screening in Jordan. These studies should investigate genetic and environmental factors underlying the high incidence of congenital heart disease in this population to shed light on possible targeted interventions.

Our study findings could be incorporated into national guidelines on healthcare; Jordanian institutions, including the Ministry of Health, should consider mandatory pulse oximetry screening. This will help in early detection of CCHD in infants, potentially reducing morbidity and mortality, and help in the achievement of SDGs 3.2 and 3.4 [20]. Regular training sessions and workshops should be conducted for health professionals regarding the importance and proper conduct of POS. Moreover, public awareness regarding early detection of CCHD can be promoted through different media forums. Further implementation science studies are needed to determine the optimal timing and methodology of training as well as the most effective strategies to introduce screening protocols to families

Screening in LLMICs is fraught with problems. Insufficiency of echocardiography because of limited resources, unavailability of trained personnel, and logistical issues could thwart the large-scale implementation of effective screening strategies. Overcoming these barriers will be essential to improving early detection and intervention outcomes for congenital heart diseases, especially as programs extend beyond referral hospitals. The introduction of Point-of-Care Ultrasound and simple ultraportable devices offers a promising solution to some of these challenges. These advancements may better facilitate screening in resource-limited settings by offering affordable and accessible diagnostic tools.

Development and execution of national guidelines/protocols for POS, including uniform procedures for screening, follow-up, and referral related to specialized care, accompanied by measures to ensure access to CCHD screening and care at an equitable level in all parts of the country, including areas that are rural and underserved, will be integral as this program expands.

Our study has several limitations. First, Al Bashir does not perform neonatal cardiac surgery, and patients are transferred to another hospital for their surgical interventions. We therefore do not have data regarding long-term outcomes, but we hope to collect these as part of future research. Additionally, we do not have data regarding false negatives for screening (those who passed CCHD screening but ultimately had CCHD), making the calculation of sensitivity in this context impossible. Furthermore, some neonates were discharged within four hours of birth against medical advice, which may have led to missed screening opportunities. Preterm infants who were admitted directly to the NICU were also excluded from screening results, potentially underestimating the true burden of CCHD in this cohort. Future studies will examine implementation of this protocol in the NICU. Finally, we did not collect qualitative data from families regarding their screening experience, which will be the subject of future research. However, no family refused screening.

## 5. Conclusions

POS for CCHD was successfully implemented at a large hospital in Jordan. This provides evidence for detection of CCHD using POS and the integration in routine neonatal care in Jordan. We found very high rates of CCHD, possibly secondary to unique sociodemographic and genetic factors. These high rates emphasize the importance of early identification. We identified a higher false-positive rate and lower specificity than previous studies, likely secondary to the need to screen many infants prior to 24 h of life. The findings support integrating the use of POS into Jordan’s national neonatal screening program.

## Figures and Tables

**Figure 1 pediatrrep-17-00023-f001:**
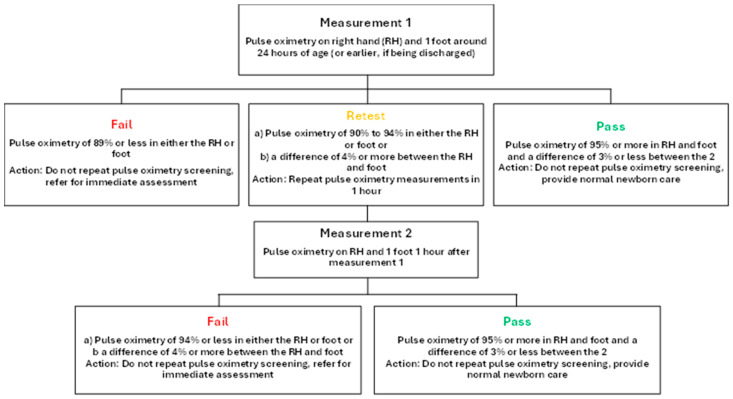
Pulse oximetry protocol. All infants were screened at 24 h of life or earlier if being discharged. Infants who failed screening were referred for echocardiogram and additional evaluation [23,24].

**Figure 2 pediatrrep-17-00023-f002:**
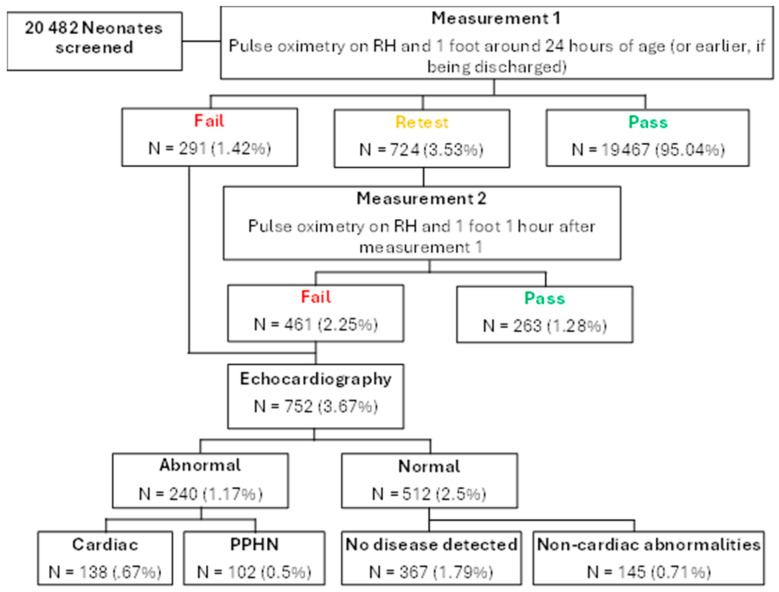
Results of screening. A total of 752 (3.7%) infants failed the screening protocol. Among these infants, echocardiography identified an abnormality in 240 (31.9%). Additional non-cardiac abnormalities, including sepsis and pneumonia, were identified in 145 (19.3%) infants.

**Table 1 pediatrrep-17-00023-t001:** Demographics of neonates screened.

	*n*	%
**Birth year**		
2022	7958	38.9
2023	8872	43.3
2024	3652	17.9
**Sex**		
Male	11,883	58.0
Female	8599	42.0
**Consanguinity** (yes)	6207	30.3
**Nationality**		
Jordanian	18,404	89.9
Syrian	1640	8.0
Palestinian	311	1.5
Iraqi	44	0.2
Yemeni	38	0.2
Sudanese	37	0.2
Pakistani	7	0.0
Ethiopian	1	0.0
	**Mean**	**SD**
**Birth body weight (Kg)**	3.2	0.7
**Gestational age (weeks)**	38.7	1.5

**Table 2 pediatrrep-17-00023-t002:** Echocardiography findings in neonates who failed screening.

	*n*	%
Normal *	512	68.1
Persistent pulmonary hypertension of the newborn *	102	13.6
Hypoplastic Left Heart Syndrome	23	3.1
Tetralogy of Fallot	18	2.4
Transposition Of Great Arteries	17	2.3
Single Ventricle	16	2.1
Double outlet right ventricle	12	1.6
Pulmonary Atresia	12	1.6
Critical pulmonary stenosis	11	1.5
Coarctation of Aorta	9	1.2
Total anomalous Pulmonary venous Return	9	1.2
Truncus Arteriosus	5	0.7
Cardiac mass *	2	0.3
Tricuspid Atresia	2	0.3
Restrictive cardiomyopathy *	1	0.1
Sub-valvular aortic stenosis	1	0.1

* Not included in CCHD definition and calculations.

**Table 3 pediatrrep-17-00023-t003:** False-positive rates by time of screening.

	Total Neonates	False Positives	*p* Value
	***n* (%)**	***n* (%)**	
**≥24 h**	7259 (35.4)	57 (0.8)	<0.001
**<24 h**	13,223 (64.6)	310 (2.3)	

## Data Availability

The raw data supporting the conclusions of this article will be made available by the authors on request.

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
