# Peer review of "Prospective Evaluation of Pulse Oximetry Screening for Critical Congenital Heart Disease in a Jordanian Tertiary Hospital: High Incidence and Early Detection Challenges"

_pediatrrep, 2025, doi:10.3390/pediatric17010023_

Round 1
Reviewer 1 Report
Comments and Suggestions for Authors
The manuscript titled Pulse Oximetry and Critical Congenital Heart Disease Screening in a Jordanian Tertiary Hospital: A Single Center Experience covers an important public health issue in a single center study; I have the following comments:
The current title is informative but could be more precise. To better reflect the study's prospective cohort design and key findings, consider: "Prospective Evaluation of Pulse Oximetry Screening for Critical Congenital Heart Disease in a Jordanian Tertiary Hospital: High Incidence and Early Detection Challenges." This revision clarifies that the study presents new data rather than a general review. Including the study design ("Prospective Cohort Study") improves transparency while highlighting key findings ("High Incidence of CCHD") and sets clear expectations for the reader. A more specific title enhances impact and readability.
Abstract (Lines 41–43): The false positive rate is reported as 1.8%, higher among infants screened before 24 hours of life (2.34%) than those screened later (0.79%, p<0.001). However, confidence intervals (CIs) are missing, making it difficult to assess the precision of these estimates. Including CIs would improve statistical rigor.
Conclusion (Lines 46–47): The statement "This highlights the importance of implementing national screening protocols to improve early diagnosis and intervention." lacks actionable recommendations. Policymakers need guidance on feasibility, cost, and integration into existing healthcare systems to make informed decisions. Addressing these aspects would enhance the manuscript's practical relevance.
National Screening Policy (Lines 79–80): The manuscript states that "Implementation in low- and low-middle-income countries (LLMIC) has been harder, delaying CCHD diagnosis and treatment." However, it does not clarify whether Jordan currently has a national CCHD screening policy. The study should compare its findings to existing protocols if a policy exists. If not, explicitly stating this would strengthen the study's justification.
Exclusion of NICU-Admitted Infants (Lines 106–107): NICU-admitted infants were excluded due to the impact of oxygen therapy on pulse oximetry screening (POS) results. However, as high-risk patients, their exclusion may underestimate the actual burden of CCHD. Other studies adjust for oxygen therapy rather than entirely exclude NICU cases. Discussing this limitation and considering alternative approaches would strengthen the study's conclusions.
Author Response
The manuscript titled Pulse Oximetry and Critical Congenital Heart Disease Screening in a Jordanian Tertiary Hospital: A Single Center Experience covers an important public health issue in a single center study; I have the following comments:
The current title is informative but could be more precise. To better reflect the study's prospective cohort design and key findings, consider: "Prospective Evaluation of Pulse Oximetry Screening for Critical Congenital Heart Disease in a Jordanian Tertiary Hospital: High Incidence and Early Detection Challenges." This revision clarifies that the study presents new data rather than a general review. Including the study design ("Prospective Cohort Study") improves transparency while highlighting key findings ("High Incidence of CCHD") and sets clear expectations for the reader. A more specific title enhances impact and readability.
We appreciate the opportunity to improve the title. We have made this revision as requested.
Abstract (Lines 41–43): The false positive rate is reported as 1.8%, higher among infants screened before 24 hours of life (2.34%) than those screened later (0.79%, p<0.001). However, confidence intervals (CIs) are missing, making it difficult to assess the precision of these estimates. Including CIs would improve statistical rigor.
We agree that providing confidence intervals makes the assessment of these data easier. We have added them as requested.
Conclusion (Lines 46–47): The statement "This highlights the importance of implementing national screening protocols to improve early diagnosis and intervention." lacks actionable recommendations. Policymakers need guidance on feasibility, cost, and integration into existing healthcare systems to make informed decisions. Addressing these aspects would enhance the manuscript's practical relevance.
While we agree that guidance regarding feasibility, cost, and integration into the existing healthcare system is important, we are not able to address these topics with the current data. We do plan to assess feasibility and cost in future studies. We have added this to the Conclusion.
National Screening Policy (Lines 79–80): The manuscript states that "Implementation in low- and low-middle-income countries (LLMIC) has been harder, delaying CCHD diagnosis and treatment." However, it does not clarify whether Jordan currently has a national CCHD screening policy. The study should compare its findings to existing protocols if a policy exists. If not, explicitly stating this would strengthen the study's justification.
There is currently no national CCHD screening policy in Jordan. We have stated this in the Introduction as suggested.
Exclusion of NICU-Admitted Infants (Lines 106–107): NICU-admitted infants were excluded due to the impact of oxygen therapy on pulse oximetry screening (POS) results. However, as high-risk patients, their exclusion may underestimate the actual burden of CCHD. Other studies adjust for oxygen therapy rather than entirely exclude NICU cases. Discussing this limitation and considering alternative approaches would strengthen the study's conclusions.
We agree that this methodological choice, which was to focus on healthy neonates admitted to the nursery, limits the application in the NICU setting. This would be an important next step in our research program. We have added discussion of this to our Limitations section. Lines 347-349 of the Limitations also acknowledge that this may lead to underestimation of CCHD burden.
Reviewer 2 Report
Comments and Suggestions for Authors
I thank the authors for the opportunity they have given me to read this interesting paper.
I would like to make a few small recommendations.
The numerical data that appear in the tables are repeated in the text. In the results section, the numerical results that appear in the tables should be removed from the text, or if the text is left, the table should be removed (for example, table 3<9).
In the numerical data, the authors sometimes use one decimal and sometimes two, without any criteria that I appreciate. The authors should unify the figures, either one or two decimals.
The conclusions should reflect only the findings reflected in the results. The authors' considerations, which although true, do not reflect the results, such as the last sentence, should be removed.
Author Response
I thank the authors for the opportunity they have given me to read this interesting paper. I would like to make a few small recommendations.
The numerical data that appear in the tables are repeated in the text. In the results section, the numerical results that appear in the tables should be removed from the text, or if the text is left, the table should be removed (for example, table 3<9).
We appreciate this feedback. The data presented in Table 3 was entirely contained in the text, and we have deleted Table 3 as suggested.
In the numerical data, the authors sometimes use one decimal and sometimes two, without any criteria that I appreciate. The authors should unify the figures, either one or two decimals.
We appreciate this comment. We have updated the report so that all numerical data have one decimal.
The conclusions should reflect only the findings reflected in the results. The authors' considerations, which although true, do not reflect the results, such as the last sentence, should be removed.
We appreciate that the last sentence is a perspective and is not directly supported by the results in the manuscript. We have deleted this sentence as suggested.
Reviewer 3 Report
Comments and Suggestions for Authors
You should introduce the "pulse oximetry" in the Introduction of the Abstract
line 92: "This study is the first performance report of its kind in Jordan, conducted to assess 92 the reliability and accuracy of using POS to screen neonates for CCHD" >> What is new in this analysis? If other studies have already proven its effectiveness
line 93-95 >> rewrite. confused
Like in the Abstract, you should intro the "Pulse oximetry". Briefly refer to the technique, operationalization and contexts.
Congratulations on the very well written, clear and concise text. The entire analysis was well carried out overall, and the minor good point is the originality and novelty of the study, where POS is a widely studied technique that should already be applied. However, I understand that the results of this study could boost its implementation in healthcare
Author Response
You should introduce the "pulse oximetry" in the Introduction of the Abstract
We have added text regarding pulse oximetry and screening to the Introduction of the Abstract as requested.
line 92: "This study is the first performance report of its kind in Jordan, conducted to assess 92 the reliability and accuracy of using POS to screen neonates for CCHD" >> What is new in this analysis? If other studies have already proven its effectiveness
Although pulse oximetry screening for CCHD is widely accepted in high-income countries, implementation in low- and middle-income counties has been limited. There are very limited reports in these setting, and there have been concerns about feasibility. We believe that our report both builds the case for implementation in middle-income countries broadly and also helps build the case in Jordan specifically.
line 93-95 >> rewrite. Confused
We agree this sentence was confusing. We had added this sentence to distinguish the neonates in this study from those identified in our previous work. We have rewritten the sentence in order to make this reasoning clearer.
Like in the Abstract, you should intro the "Pulse oximetry". Briefly refer to the technique, operationalization and contexts.
We appreciate that pulse oximetry may not be familiar to some readers. We have added text to the Introduction to describe the technology and contexts in which it is used.
Congratulations on the very well written, clear and concise text. The entire analysis was well carried out overall, and the minor good point is the originality and novelty of the study, where POS is a widely studied technique that should already be applied. However, I understand that the results of this study could boost its implementation in healthcare
We appreciate this positive feedback on our manuscript. As stated above, we agree that the efficacy of pulse oximetry screening for CCHD has been established. However, we believe the presented manuscript contributes substantially to the literature regarding implementation of CCHD screening outside high-income countries.